# Treatment effects beyond the mean using distributional regression: Methods and guidance

**Maike Hohberg** [ORCID]*, **Peter Pütz, Thomas Kneib**

Chair of Statistics, Faculty of Economics, University of Goettingen, Goettingen, Germany

* mhohber@uni-goettingen.de

## Abstract

This paper introduces distributional regression also known as generalized additive models for location, scale and shape (GAMLSS) as a modeling framework for analyzing treatment effects beyond the mean. In contrast to mean regression models, GAMLSS relate each distributional parameter to covariates. Therefore, they can be used to model the treatment effect not only on the mean but on the whole conditional distribution. Since they encompass a wide range of different distributions, GAMLSS provide a flexible framework for modeling non-normal outcomes in which additionally nonlinear and spatial effects can easily be incorporated. We elaborate on the combination of GAMLSS with program evaluation methods including randomized controlled trials, panel data techniques, difference in differences, instrumental variables, and regression discontinuity design. We provide practical guidance on the usage of GAMLSS by reanalyzing data from the Mexican Progresa program. Contrary to expectations, no significant effects of a cash transfer on the conditional consumption inequality level between treatment and control group are found.

## 1 Introduction

Program evaluation typically identifies the effect of a policy or a program on the mean of the response variable of interest. This effect is estimated as the average difference between treatment and comparison group with respect to the response variable, potentially controlling for confounding covariates. However, questions such as "How does the treatment influence a person's future income distribution" or "How does the treatment affect consumption inequality conditional on covariates" cannot be adequately answered when evaluating mean effects alone. Concentrating on mean differences between a treatment group and a comparison group is likely to miss important information about changes along the whole distribution of an outcome, for example in terms of an unintended increase in inequality, or when targeting ex ante vulnerability to a certain risk. These are concepts that do not only take the mean into account but rely on other measures such as the variance and skewness of the response.

As shown by Bitler et al. [1], analyzing average effects in subgroups does not adequately capture heterogeneities along the outcome distribution. For a systematic and coherent analysis

**Data Availability Statement:** URLs to access the data and software code are within the manuscript and Supporting Information files.

**Funding:** MH received funding from the Ministry for Science and Culture of Lower Saxony within the project "Reducing Poverty Risks in Developing

Countries" and the German Research Foundation (DFG) within the research project KN 922/9-1 "Semiparametric Regression Models for Location, Scale and Shape". PP received funding from the German Research Foundation (DFG) within the Collaborative Research Center "Ecological and Socioeconomic Functions of Tropical Lowland Rainforest Transformation Systems (Sumatra, Indonesia)". The funders did not play any role in the study design, data collection and analysis, decision to publish, or preparation of the manuscript.

**Competing interests:** The authors have declared that no competing interests exist.

of treatment effects on all functionals of the response distribution, we introduce generalized additive models for location, scale and shape (GAMLSS, [2]) to the evaluation literature. GAMLSS allow all parameters of the response distribution to vary with explanatory variables and can hence be used to assess how the conditional response distribution changes due to the treatment. In addition, GAMLSS constitute an overarching framework to easily incorporate nonlinear, random, and spatial effects and hence to flexibly model the relationship between the covariates and the predictors. The method encompasses a wide range of potential outcome distributions, including discrete and multivariate distributions, and distributions for shares. Due to estimating only *one* model including all distributional parameters, practically every distribution functional (quantiles, inequality measures such as the Gini coefficient, etc.) can be derived consistently from the conditional distribution making the scope of application manifold.

Besides a brief review of the methodological background for GAMLSS, our main contributions are (i) to link the flexibility of GAMLSS modelling with treatment effect evaluation and (ii) to practically demonstrate how to conduct such a treatment effect evaluation. We specifically highlight the additional information that can be drawn from treatment effects beyond the mean. For this purpose, we have chosen an example that is very familiar to the evaluation community in economics: We rely on the same household survey used in Angelucci and De Giorgi [3] to evaluate Progresa/Oportunidades/Prospera—a cash transfer program in Mexico. Initiated in 1997, the experimental design of the program allocated cash transfers to poor families in treatment villages in exchange for the households' children regularly attending school and for utilizing preventive care measures regarding health and nutrition. By using this extensively researched program as our application example, we show additional results using GAMLSS. In fact, we find no significant decline in food consumption inequality after the introduction of conditional cash transfers—a result that has gone unnoticed in the several analyses of the program's heterogeneous effects (e.g., [4, 5]).

While GAMLSS have not been used in the context of program evaluation, there is a substantial strand of literature that focuses on treatment effects on the whole distribution of an outcome or, to put it differently, on building counterfactual distributions. The idea is to consider the distribution of the treated versus their distribution if they had not been treated. The literature generally differentiates between effects on the unconditional distribution and the conditional distribution. While the effects on the unconditional distribution and unconditional quantile effects have been dealt with (e.g., [6–10]), the focus of this paper is the conditional distribution and the functionals that can be derived from it. Conditional distributions are of interest, when analyzing the effect heterogeneity based on the observed characteristics [10]. Especially in the case of inequality, conditional distributions are important to differentiate between within and between variance. For example, differences in consumption or income might stem from different characteristics or abilities such as years of education. With conditional distributions, we, however, assess the differences in consumption or income for individuals with equal or similar education and work experience. The fair notion would be that a person with higher education and more work experience earns more. It is the conditional inequality that is perceived as unfair.

To estimate the conditional distribution, a popular approach is to use quantile regression [11, 12]. Quantile regression is a very powerful instrument if one is interested in the effect at a specific quantile but distributional characteristics can only be derived after the effects at a very high number of quantiles have been estimated yielding an approximation of the whole distribution. As we believe that quantile regression is most familiar to practitioners when estimating effects beyond the mean, we will elaborate a direct comparison of GAMLSS and quantile regression in Section 3.

Other interesting approaches to go beyond the mean in regression modeling include Chernozhukov et al. [13] and Chernozhukov et al. [14] who introduce "distribution regression". Building upon Foresi and Peracchi [15], they develop models that do not assume a parametric distribution but estimate the whole conditional distribution flexibly. The basic idea is to estimate the distribution of the dependent variable via several binary regressions for $F(z|x_i) = Pr(y_i \leq z|x_i)$ based on a fine grid of values $z$. These models have the advantage of not requiring an assumption about the form of the response distribution. However, they require constrained estimates to avoid crossing predictions similar to crossing quantiles in quantile regression. Recently, Shen [16] proposed a nonparametric approach based on kernel functions to estimate the effect of minimum wages on the conditional income distribution. She points out that the flexibility of estimating distributional effects conditional on the other covariates is also useful for the regression discontinuity design (RDD). In Shen and Zhang [17] they develop tests relating the stochastic dominance testing to the RDD.

Thus, different concepts are already introduced with different scope for application. By applying GAMLSS to the evaluation context, we provide a flexible, parametric complement to the existing approaches. The advantage of this approach is that it provides *one* coherent model for the conditional distribution which estimates simultaneously the effect on all distributional parameters avoiding crossing quantiles or crossing predictions. If the distributional assumption is appropriate, the parametric approach allows us to rely on classical results for inference in either frequentist or Bayesian formulations, including large sample theory. The parametric formulation furthermore enables us to derive various quantities of interest from the same estimated distribution (quantiles, moments, Gini coefficient, interquartile range, etc.) which are all consistent with each other. As the distributional assumption obviously plays a crucial role in GAMLSS, we suggest guiding steps and easy-to-use tools for the practitioner to decide on a distribution.

The remainder is structured as follows: Section 2 provides the methodological background of GAMLSS. Section 3 elaborates on the potential benefits and limitations of GAMLSS for evaluating treatment effects. A practical step-by-step implementation and interpretation is given in Section 4. Though this section uses data from a randomized controlled trial (RCT), the methodology proposed in this paper applies to non-experimental methods as well. The appendix elaborates on the combination of GAMLSS with other evaluation methods including panel data approaches, difference-in-differences, instrumental variables (IV), and regression discontinuity design (RDD). Section 5 concludes.

## 2 Generalized additive models for location, scale and shape

### 2.1 A general introduction to GAMLSS

For the sake of illustration, we start with a basic regression as it would be used, for example, when evaluating data from an RCT. Based on observed values $(\mathbf{x}'_i, T_i, y_i)$, $i = 1, \ldots, n$, we are interested in determining the regression relation between a treatment, $T_i$, and the response variable $y_i$, while controlling for a vector of non-stochastic covariates $\mathbf{x}'_i$. For simplicity and in line with the application in Section 4, we describe the method in the context of a binary treatment but it applies to the continuous case as well. A corresponding simple linear model

$$y_i = \beta_0 + \beta_T T_i + \mathbf{x}'_i \boldsymbol{\beta}_1 + \varepsilon_i \qquad (1)$$

with error terms $\varepsilon_i$ subject to $E(\varepsilon_i) = 0$ implies that the treatment and the remaining covariates linearly determine the expectation of the response via

$$E(y_i) = \mu_i = \beta_0 + \beta_T T_i + \mathbf{x}'_i \boldsymbol{\beta}_1. \qquad (2)$$

If, in addition, the distribution of the error term is assumed to not functionally depend on the observed explanatory variables (implying, for example, homoscedasticity), the model focuses exclusively on the expected value, that is, it is a mean regression model. In other words, all effects that do not affect the mean but other parameters of the response distribution such as the scale parameter are implicitly subsumed into the error term.

One possibility to weaken the focus on the mean and give more structure to the remaining effects is to relate all parameters of a response distribution to explanatory variables. In the case of a normally distributed response $y_i \sim N(\mu_i, \sigma_i^2)$, both mean and variance could depend on the explanatory variables. Assuming again one treatment variable $T_i$ and additional covariates $\mathbf{x}_i'$, the corresponding relations in a GAMLSS can be specified as follows:

$$\mu_i = \beta_0^\mu + \beta_T^\mu T_i + \mathbf{x}_i' \boldsymbol{\beta}_1^\mu, \tag{3}$$

$$\log(\sigma_i) = \beta_0^\sigma + \beta_T^\sigma T_i + \mathbf{x}_i' \boldsymbol{\beta}_1^\sigma. \tag{4}$$

Here, the superscripts in $\beta_0^\mu, \beta_T^\mu, \boldsymbol{\beta}_1^\mu, \beta_0^\sigma, \beta_T^\sigma$ and $\boldsymbol{\beta}_1^\sigma$ indicate the dependency of the intercepts and slopes on the respective distribution parameters. The log transformation in Eq (4) is applied in order to guarantee positive standard deviations for any value of the explanatory variable.

Aside from the normal distribution, a wide range of possible distributions is incorporated in the flexible GAMLSS framework (see e.g. [18] for a comprehensive overview of distributions used with GAMLSS):

(a). In addition to distributions with location and scale parameters, distributions with skewness and kurtosis parameters can be modeled to account for regression effects on such features.

(b). For count data, not only the Poisson but also alternative distributions such as the negative binomial distribution that accounts for over-dispersion or compound distributions accounting for zero-inflation can be used. Outside the GAMLSS context, these distributions are for example used in Chen et al. [19] for crash frequency modeling.

(c). Often we consider nonnegative dependent variables, e.g. income, with an amount of zeros that cannot be captured by continuous distributions. For these cases, a mixed discrete-continuous distribution can be used that combines a nonnegative continuous distribution with a point mass in zero.

(d). For response variables that are shares, also called fractional responses, we can consider continuous distributions defined on the unit interval.

(e). Even multivariate distributions can be placed within this modeling framework (see e.g. [20]).

GAMLSS assume that the observed $y_i$ are conditionally independent and that their distribution can be described by a parametric density $p(y_i|\vartheta_{i1}, \ldots, \vartheta_{iK})$ where $\vartheta_{i1}, \ldots, \vartheta_{iK}$ are $K$ different parameters of the distribution. For each of these parameters we can specify an equation of the form

$$g_k(\vartheta_{ik}) = \beta_0^{\vartheta_k} + \beta_T^{\vartheta_k} T_i + x_i' \boldsymbol{\beta}^{\vartheta_k}, \tag{5}$$

where the link function $g_k$ ensures the compliance with the requirements of the parameter space, such as the log link to ensure positive variances in Eq (4). Linking the parameters to an unconstrained domain also facilitates the consideration of semiparametric, additive regression

specifications including, for example, nonlinear, spatial or random effects. Due to assuming a distribution for the response variable, the likelihood is readily available such that model estimation can be conducted for example by (penalized) maximum likelihood [2] or in a Bayesian framework based on Markov chain Monte Carlo simulations [21, 22].

## 2.2 Additive predictors

The univariate case described in the previous subsection can be easily extended to a multivariate and even more flexible setting. In particular, each parameter $\vartheta_{ik}$, $k = 1, \ldots, K$, of the response distribution is now conditioned on several explanatory variables and can be related to a predictor $\eta_i^{\vartheta_k}$ via a link function $g_k$ such that $\vartheta_{ik} = g_k^{-1}(\eta_i^{\vartheta_k})$.

A generic predictor for parameter $\vartheta_{ik}$ takes on the following form:

$$\eta_i^{\vartheta_k} = \beta_0^{\vartheta_k} + \beta_T^{\vartheta_k} T_i + f_1^{\vartheta_k}(\mathbf{x}_{1i}) + \cdots + f_{J_k}^{\vartheta_k}(\mathbf{x}_{J_k i}). \tag{6}$$

This representation shows nicely why we refer to $\eta_i^{\vartheta_k}$ as a "structured additive predictor". While $\beta_0^{\vartheta_k}$ denotes the overall level of the predictor and $\beta_T^{\vartheta_k}$ is the effect of a binary treatment on the predictor, functions $f_j^{\vartheta_k}(\mathbf{x}_{ji})$, $j = 1, \ldots, J_k$, can be chosen to model a range of different effects of a vector of explanatory variables $\mathbf{x}_{ji}$:

(a).  Linear effects are captured by linear functions $f_j^{\vartheta_k}(\mathbf{x}_{ji}) = x_{ji}\beta_j^{\vartheta_k}$, where $x_{ji}$ is a scalar and $\beta_j^{\vartheta_k}$ a regression coefficient.

(b).  Nonlinear effects can be included for continuous explanatory variables via smooth functions $f_j^{\vartheta_k}(\mathbf{x}_{ji}) = f_j^{\vartheta_k}(x_{ji})$ where $x_{ji}$ is a scalar. We recommend using P(enalized)-splines [23] due to their versatility in approximating even complex nonlinear effects.

(c).  Accounting for spatial autocorrelation is a specific challenge due to the resulting spatial dependence between the observed response variables. In GAMLSS, this problem is dealt with by including a spatially correlated random effect in one or multiple of the regression predictors (similar as in Zeng et al. [24] for an ordinal logit model) where the exact specification depends on the type of spatial information. If the spatial allocation is given in terms of geographical coordinates, the spatial pattern can be accounted for via Gaussian random fields $f_j^{\vartheta_k}(\mathbf{x}_{ji}) = f_j^{\vartheta_k}(s_{x,i}, s_{y,i})$, where an appropriate covariance function is utilized to determine the dependence between observations and $s_{x,i}, s_{y,i}$ are the coordinates of observation $i$. If spatial information is given in terms of administrative units $s_i$, Gaussian Markov random fields enable the specification of spatial dependence based on the neighborhood structure of the regions, see [25] for a detailed discussion of both options.

(d).  If the data are clustered, random or fixed effects $f_j^{\vartheta_k}(\mathbf{x}_{ji}) = \beta_{j,g_i}^{\vartheta_k}$ can be included to adjust for unobserved, group-specific heterogeneity as well as within-group dependence, where $g_i$ denoting the cluster the observations are grouped into.

Consequently, GAMLSS allows the researcher to incorporate very different types of effects within one modeling framework. Estimation may then be done via a back-fitting approach within the Newton-Raphson type algorithm that maximizes the penalized likelihood and estimates the unknown quantities simultaneously. The methodology is implemented in the `gamlss` package in the software R, and described extensively in Stasinopoulos and Rigby [26] and Stasinopoulos et al. [18]. Alternatively, a Bayesian implementation is available in the open source software `BayesX` [27, 28].

## 2.3 GAMLSS vs. quantile regression

A popular alternative to simple mean regression is quantile regression, see, for example, [12] for an excellent introduction. Quantile regression relates not the mean but quantiles of the outcome variable to explanatory variables without making a distributional assumption about the outcome variable. In addition to requiring independence of observed values $y_i$, a quantile regression model with one explanatory variable $x_i$ only assumes that

$$y_i = \beta_{0,\tau} + \beta_{1,\tau} x_i + \varepsilon_{i,\tau} \tag{7}$$

where $\varepsilon_{i,\tau}$ is a quantile-specific error term with the quantile condition $P(\varepsilon_{i,\tau} \leq 0) = \tau$ replacing the usual assumption $E(\varepsilon_{i,\tau}) = 0$. This implies a specific form of the relationship: The explanatory variable influences the $\tau$-quantile in a linear fashion. Thus, the model can still be misspecified even though we do not make an assumption about the distribution of the response. A further disadvantage of quantile regression is that the response variable must be continuous. This is especially problematic in the case of discrete or binary data, continuous distributions with a probability greater than zero for certain values or when the dependent variable is a proportion. This is different to the GAMLSS approach that also includes those cases. Note that we appraise GAMLSS as a generic framework here, even though it does not yield additional benefits if the distribution has only one parameter such as the binomial or Poisson distribution. Another problem in quantile regression is the issue of crossing quantiles [29]. Theoretically, quantiles should be monotonically ordered according to their level such that $\beta_{0,\tau_1} + \beta_{1,\tau_1} x_i \leq \beta_{0,\tau_2} + \beta_{1,\tau_2} x_i$ for $\tau_1 \leq \tau_2$ and all $x_i$, $i = 1, \ldots, n$. Since the regression models are estimated for each quantile separately, this ordering does not automatically enter the model and crossing quantiles can occur especially when the amount of considered quantiles is large in order to approximate the whole distribution. If one assumes parallel regression lines, crossing quantiles can be avoided. However, in this case the application of quantile regression becomes redundant since for each quantile only the intercept parameter shifts while the effect of the explanatory variables would be independent from the quantile level. Therefore, the models rely on the less restrictive assumption that quantiles should not cross for the observed values of the explanatory variables. Strategies to avoid quantile crossing include simultaneous estimation, for example, based on a location scale shift model [30], on spline based non-crossing constraints [31], or on quantiles sheets [32]. Chernozhukov et al. [33] and Dette and Volgushev [34] propose estimating the conditional distribution function first and inverting it to obtain quantiles. However, all of these alternatives require additional steps and most of them cannot easily incorporate an additive structure for the predictors [35]. In empirical research, conventional quantile regression is predominantly used by far. In any case, quantile regression estimates the relationship for certain quantiles separately but does not have a model to estimate the complete distribution. This can be also problematic if measures other than the quantiles such as the standard deviation or Gini coefficient should be analyzed.

In contrast, GAMLSS are consistent models from which any feature of a distribution can be derived. If the assumed distribution is appropriate, GAMLSS can provide more precise estimators than quantile regression especially for the tails of the empirical distribution where data points are scarce. Since we use maximum likelihood for estimation, a variety of related methods and inference techniques that rely on the distributional assumption can be used such as likelihood ratio tests and confidence intervals. As simulation studies in Klein et al. [36] show bad performance for likelihood-based confidence intervals in certain situations, we will, however, rely on bootstrap inference for the application in Section 4. The main drawback of GAMLSS is a potential misspecification but Section 4 presents associated model diagnostics to minimize this risk. Besides the methodological differences, quantile regression and GAMLSS

expose their benefits in different contexts. Following Kneib [35], we suggest using quantile regression if the interest is on a certain quantile of the distribution of the dependent variable. On the other hand, the GAMLSS framework is more appropriate if one is interested in the changes of the entire conditional distribution, its parameters and certain distributional measures relying on these parameters, such as the Gini coefficient.

## 3 Potentials and pitfalls of GAMLSS for analyzing treatment effects beyond the mean

GAMLSS can be applied to evaluation questions when the outcome of interest is not the difference in the expected mean of treatment and comparison group but the whole distribution and derived distributional measures while at the same accounting for differences in covariates. Compared to an analysis where the distributional measures are themselves the dependent variable, the great advantage of GAMLSS is that they yield *one* model from which several measures of interest can be coherently derived. In case of income, for example, these measures might be expected income, quantiles, Gini, the risk of being poor etc. Thereby, consistent results are obtained since all measures are based on the same model using the same data. Furthermore, aggregated distributional measures as dependent variables mask the underlying individual information. On the contrary, GAMLSS allows the researcher to estimate (treatment) effects on aggregate measures on the individual level.

To present some examples of beyond-the-mean-measures, we focus in the following on inequality and vulnerability to poverty but a lot more measures can be analyzed using GAMLSS. For example, as Meager [37] points out, risk profiles of business profits which are important for the functioning of the credit market are based on characteristics of the entire distribution and not only the mean.

### 3.1 Example: GAMLSS and vulnerability as expected poverty

Ex ante poverty measures such as vulnerability to poverty are an interesting outcome if one is not only interested in the current (static) state of poverty but also in the probability of being poor. Although there are different concepts of vulnerability, see [38] for an overview and empirical comparison of different vulnerability measures, we focus on the notion of vulnerability as expected poverty [39]. In this sense, vulnerability is the probability of having a consumption (or income) level below a certain threshold. To calculate this probability, separate regressions for mean and variance of log consumption are traditionally estimated using the feasible generalized least squares estimator (FGLS, [40]), yielding an estimate for the expected mean and variance for each household. Concretely, the procedure involves a consumption model of the form

$$\ln y_i = \beta_0^\mu + \mathbf{x}_i' \boldsymbol{\beta}_1^\mu + \varepsilon_i, \tag{8}$$

where $y_i$ is consumption or income, $\beta_0$ an intercept, $\mathbf{x}_i$ is a vector of household characteristics, $\boldsymbol{\beta}_1$ is a vector of coefficients of the same length and $\varepsilon_i$ is a normally distributed error term with variance

$$\sigma_{e,i}^2 = \beta_0^\sigma + \mathbf{x}_i' \boldsymbol{\beta}_1^\sigma. \tag{9}$$

To estimate the intercepts $\beta_0^\mu$ and $\beta_0^\sigma$ and the vectors of coefficients $\boldsymbol{\beta}_1$ and $\boldsymbol{\beta}_1^\sigma$ the 3-step FGLS procedure involves several OLS estimation and weighting steps. Assuming normally distributed log incomes $\ln y_i$, the estimated coefficients are plugged into the standard normal

cumulative distribution function

$$\widehat{\Pr}(\ln y_i < \ln z | \mathbf{x}_i') = \Phi\left(\frac{\ln z - (\hat{\beta}_0^\mu + \mathbf{x}_i'\hat{\boldsymbol{\beta}}_1^\mu)}{\sqrt{\hat{\beta}_0^\sigma + \mathbf{x}_i'\hat{\boldsymbol{\beta}}_1^\sigma}}\right), \qquad (10)$$

where $\hat{\beta}_0^\mu + \mathbf{x}_i'\hat{\boldsymbol{\beta}}_1^\mu$ is the estimated mean, $\sqrt{\hat{\beta}_0^\sigma + \mathbf{x}_i'\hat{\boldsymbol{\beta}}_1^\sigma}$ the estimated standard deviation, and $z$ the poverty threshold. A household is typically classified as vulnerable if the probability is equal or larger than 0.5. In contrast to the 3-step FGLS procedure, GAMLSS allow us to estimate the effects on mean and variance simultaneously avoiding the multiple steps procedure. While the efficiency gain of a simultaneous estimation is not necessarily large, its main advantage is the quantification of uncertainty as it can be assessed in one model. In a stepwise procedure, each estimation step is associated with a level of uncertainty that has to be accounted for in the following step. Additionally, GAMLSS provide the flexibility to relax the normality assumption of log consumption or log income.

## 3.2 Example: GAMLSS for inequality assessment

Although inequality is normally not a targeted outcome of a welfare program, it is considered as an unintended effect since a change in inequality is likely to have welfare implications. To assess inequality, our application in Section 4 focuses on the Gini coefficient but other inequality measures are also applied. In general, we focus on the conditional distribution of consumption or income, that is, the treatment effects will be derived for a certain covariate combination. In other words, in order to analyze inequality, we do not measure unconditional inequality of consumption or income, for instance, for the entire treatment and comparison group, but inequality given that other factors that explain differences in consumption are fixed at certain values. Thus, for each combination of explanatory variables an estimated inequality measure is obtained which represents inequality unexplained by these variables. The economic reasoning is that differences in consumption or income are not *per se* welfare reducing inequality since those differences might stem from different characteristics or abilities such as years of education. We, however, assess the differences in consumption or income for those with equal or similar education as it is the conditional inequality that is perceived as unfair.

It is important to address some limitations regarding model selection and a priori model specification. As the researcher has to select explanatory variables for more than one parameter and a suitable response distribution, uncertainty in estimation can increase, yielding invalid $p$-values and possibilities for $p$-hacking open up. Note, however, that there is a trade-off between misspecification by simplifying the model via assuming constant distributional parameters and misspecifying a more complex model. Additionally, a linear regression model is certainly less complex to specify but more limited in its informative value. To reduce the chance for misclassification of more complex GAMLSS, we suggest scrutinizing the model using the criteria and tools for model diagnosis presented in Section 4. It is also common in practice to report more than one model to check robustness to model specification. Covariates can be pre-specified either on theoretical grounds or by using information from previous studies. To some extent, this is also possible for the response distribution. The type of response (continuous, nonnegative, binary, discrete etc.) already restricts the set of possible distributions to choose from. Previous studies might also give hints about the distribution of the response.

## 4 Applying GAMLSS to experimental data

### 4.1 General procedure

To demonstrate how the analysis of treatment effects can benefit from GAMLSS, we replicate and extend an evaluation study of a popular economic intervention and show how a distributional analysis could be implemented step by step.

We propose the following procedure to implement GAMLSS:

(a).  Choose potentially suitable conditional distributions for the outcome variable.

(b).  Make a (pre-)selection of covariates according to your hypothesis, theoretical considerations, etc.

(c).  Estimate your models and assess their fit, decide whether to include nonlinear, spatial, and/or random effects.

(d).  Optionally: Refine your variable selection according to statistical criteria.

(e).  Interpret the effects on the distributional parameters (if such an interpretation is available for the chosen distribution), derive the effects on the complete distribution and identify the treatment effect on related distributional measures.

In the following, we apply all of these steps to the Progresa data as used in Angelucci and De Giorgi [3] to provide a hands on guide on how to use GAMLSS in impact evaluation. The Mexican conditional cash transfer (CCT) program Progresa (first renamed Oportunidades and then Prospera) transfers money to households if they comply with certain requirements which include, for example, children's regular school attendance. CCTs have been popular development instruments over the last two decades and most researchers working in the area are well familiar with their background making it an ideal demonstration example.

### 4.2 Application: Progresa's treatment effect on the distribution

In their study "Indirect Effects of an Aid Program", Angelucci and De Giorgi [3] investigate how CCTs to targeted, eligible (poor) households affect, among other outcomes, the average food consumption of both eligible and ineligible (non-poor) households. An RCT was conducted at the village-level and information is available for four groups: eligible and ineligible households in treatment and control villages. Aside from the expected positive effect of the cash transfer on the eligible households' mean food consumption, Angelucci and De Giorgi [3] also find a considerable increase in the ineligible households' mean food consumption in the treatment villages. They link the increase to reduced savings among the non-poor, higher loans, and monetary and in-kind transfers from family and friends. Consequently, the average program effect on food consumption for the treated villages is larger than assumed when only looking at the poor. Estimating the same relationship using GAMLSS provides important information for policymakers on effects within a group, for example, whether conditional food consumption inequality decreases for an average household among the poor (or the non-poor or all households). We will assess the effect on conditional inequality via the Gini coefficient, which is defined by

$$G = \frac{\sum_{i=1}^{n}\sum_{j=1}^{n}|y_i - y_j|}{2n\sum_{h=1}^{n}y_i}, \ 0 \leq G \leq 1, \tag{11}$$

for a group of $n$ households, where $y_i$ denotes the nonnegative consumption of household $i$. For a given continuous consumption distribution function $p(y)$, which we will estimate via GAMLSS, the Gini coefficient can be written as

$$G = \frac{1}{2\mu} \int_0^\infty \int_0^\infty p(y)p(z) \, |y - z| \, dy \, dz, \tag{12}$$

with $\mu$ denoting the mean of the distribution.

Thus, a positive treatment effect on consumption in one group results in a lower Gini coefficient if all group members benefit equally, as the differences in (11) and (12) remain the same, but the denominators increase. However, there might be reasons why in one group, e.g. among the poor, only the better off benefit and the poorest do not, resulting in higher inequality.

Using GAMLSS, we investigate the program's impact on conditional food consumption inequality measured by the Gini coefficient within the non-poor and poor by comparing the treatment and control groups. In particular, we model food consumption by an appropriate distribution and link its parameters to the treatment variable and other covariates. We obtain estimates for the conditional food consumption distribution for treated and untreated households and the corresponding Gini coefficients. The pairs cluster bootstrap is applied for obtaining an inferential statement on the equality of Gini coefficients; see Section B.2 in the appendix for a description of this bootstrap method. Furthermore, we investigate the effect of Progresa on global inequality by comparing treatment and control villages, that is, all households in treatment villages are considered as treated and all households in control villages as not treated. Since the average treatment effects found by Angelucci and De Giorgi [3] are larger for the poor than for the non-poor, a lower food consumption inequality (measured by the Gini coefficient) in the treatment villages is expected. However, a higher Gini could arise if the program benefits are very unequally distributed. Decreasing inequality is an expected, even though often not explicitly mentioned and scrutinized target of poverty alleviation programs and considered to be desirable, especially in highly unequal societies such as Mexico.

Regarding the data set and specification, we refer to Table 1 in Angelucci and De Giorgi [3] and restrict our analyses to the sample collected in November 1999 and the more powerful specifications including control variables. We rely on (nearly) the same data and control variables as Angelucci and De Giorgi [3]. Along the steps described in Section 4.1, we will show in detail how to apply our modeling framework to the group of ineligibles which are also the main focus group of Angelucci and De Giorgi [3]. The corresponding software code can be downloaded from https://www.uni-goettingen.de/de/511092.html, whereas the dataset is available on https://www.aeaweb.org/articles?id=10.1257/aer.99.1.486.

**Table 1. Summary of the quantile residuals for the model based on the log-normal distribution and Singh-Maddala distribution.**

|  | Log-normal | Singh-Maddala |
|---|---|---|
| Mean | -0.000091 | -0.001102 |
| Variance | 1.000235 | 0.998379 |
| Coef. of Skewness | 0.701639 | 0.060098 |
| Coef. of Kurtosis | 6.016006 | 3.115085 |
| Filliben Correlation Coef. | 0.984499 | 0.999201 |

Notes: A good fit is indicated by values close to 0, 1, 0, 3 and 1 for mean, variance, skewness, kurtosis, and Filliben correlation coefficient, respectively.

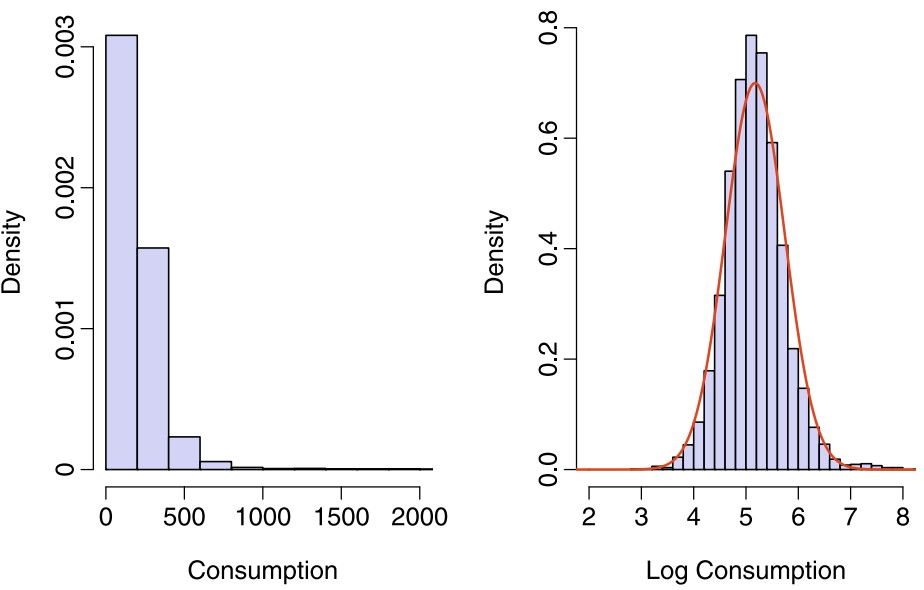

**Fig 1. Distribution of food consumption and log food consumption.**

**4.2.1 Choice of potential outcome distributions.** The distribution of the outcome variable often gives some indication about which conditional distributions are appropriate candidates. To check the adequacy of the model fit and the appropriateness of the chosen distribution, (randomized) normalized quantile residuals [41] should be used.

The histogram of the dependent variable in the left panel of Fig 1 shows a heavily right-skewed distribution while taking the logarithm yields approximately a normal distribution (right panel) such that the log-normal distribution appears to be a reasonable starting point. It has the additional advantage that it also renders easily interpretable effects of the explanatory variables on the mean and variance of the dependent variable, at least on the logarithmic scale. As a more flexible alternative, we will also consider the three-parameter Singh-Maddala that is also known as Burr Type XII distribution and capable of modeling right-skewed distributions with fat tails, see [42] for details. Note that the three parameters of the Singh-Maddala distribution do not allow a direct interpretation of effects on moments of the distribution.

**4.2.2 Preliminary choice of potentially relevant covariates.** We select the same covariates as in Angelucci and De Giorgi [3] and relate all of them to all parameters of our chosen distribution. The model contains nine explanatory variables per parameter: Besides the treatment variable, these are six variables on the household level, namely poverty index, land size, the household head's gender, age, whether they speak an indigenous language and are illiterate, as well as a poverty index and the land size as variables on the locality level. For the model relying on a log-normal distribution, two parameters $\mu$ and $\sigma$ are related to these variables,

$$\log(\mu_i) = \beta_0^\mu + T_i\beta_T^\mu + \mathbf{x}_i'\boldsymbol{\beta}_1^\mu, \tag{13}$$

$$\log(\sigma_i) = \beta_0^\sigma + T_i\beta_T^\sigma + \mathbf{x}_i'\boldsymbol{\beta}_1^\sigma, \tag{14}$$

where $T_i$ is the treatment dummy, $\beta_T^\mu$ and $\beta_T^\sigma$ are the treatment effects on the parameters $\mu$ and $\sigma$, respectively, $\mathbf{x}_i$ is a vector containing the values of the remaining covariates for household $i$ and $\boldsymbol{\beta}_1^\mu$ and $\boldsymbol{\beta}_1^\sigma$ are the corresponding coefficient vectors of the same length. In the specification relying on the three-parameter Singh-Maddala distribution, where $\mu$ and $\sigma$ are modeled as in

(13) and (14), respectively, an additional parameter $\tau$ is linked to the nine explanatory variables,

$$\log(\tau_i) = \beta_0^\tau + T_i \beta_T^\tau + \mathbf{x}_i' \boldsymbol{\beta}_1^\tau, \tag{15}$$

resulting in the considerable amount of 30 quantities to estimate as each parameter equation includes an intercept. This is, however, still a moderate number considering the sample size of more than 4,000 households in the sample of ineligibles and even less problematic for the sample of eligibles with about 10,500 observations and the combined sample. In general, if the sample size is large, it is advisable to relate all parameters of a distribution to all variables which potentially have an effect on the dependent variable and its distribution, respectively. Exceptions may include certain distributions such as the normal distribution when there are convincing theoretical arguments why a variable might affect one parameter such as the mean but not another one such as, for example, the variance. For smaller sample sizes, higher order parameters such as skewness or kurtosis parameters may be modeled in simpler fashion with few explanatory variables.

**4.2.3 Model building and diagnostics.** The adequacy of fit is assessed by some statistics of the normalized quantile residuals. As a generic tool applicable to a wider range of response distributions than deviance or Pearson residuals, these residuals were shown to follow a standard normal distribution under the true model. In Fig 2 and Table 1 it can be seen that both q-q plot and statistics reveal that the log-normal distribution might be an inadequate choice for modeling the consumption distribution. Especially the overly large coefficient of kurtosis, which should be close to 3, and the apparent skewness of the normalized quantile residuals, visible in the plot, suggest a distribution with a heavier right tail.

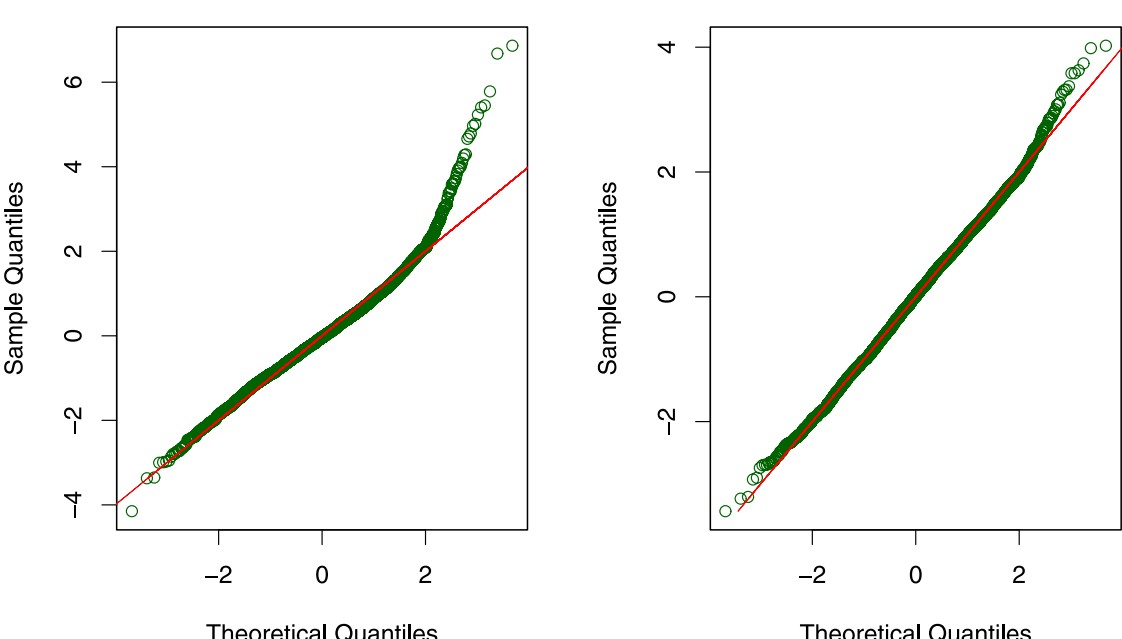

**Fig 2. Diagnosis plots for the model based on the log-normal distribution (left panel) and the Singh-Maddala distribution (right panel).**

A model relying on the Singh-Maddala distribution yields a much more satisfying diagnostic fit (see Fig 2 and Table 1). The q-q plot does not show severe deviations from the standard normal distribution, which is confirmed by the summary measures of the quantile residuals. More specifically, the Filliben correlation coefficient (measuring the correlation between theoretical and sample quantiles as displayed in the q-q plot) is almost equal to 1, the coefficient of skewness is now close to 0 and the coefficient of kurtosis close to 3. Additionally, the mean and the variance do not deviate much from their "desired" values 0 and 1, respectively.

Consequently, the Singh-Maddala distribution is an appropriate choice for modeling the present consumption data. Other diagnostic tools, as described in Stasinopoulos and Rigby [26], can be applied as well. In their application, Angelucci and De Giorgi [3] cluster the standard errors at the village level as some intra-village correlation is likely to occur. In a heuristic approach, we regress the quantile residuals of the model above on the village dummies and obtain an adjusted $R^2$ of about 10% and a very low $p$-value for the overall $F$-Test. This suggests unobserved village heterogeneity which we account for by applying a pairs cluster bootstrap procedure to obtain cluster-robust inference. Alternatively, random effects could be applied to model unexplained heterogeneity between villages. Since we use the same covariates as in Angelucci and De Giorgi [3], we do not include nonlinear covariate effects in our model specification. The model diagnostics indicate a reasonable fit and we are not particularly interested in the effects of the continuous covariates but as a check we ran a model with nonparametric covariate effects and obtained very similar results. In general, we advocate the use of nonparametric specifications for most continuous covariates.

**4.2.4 Variable selection.** A comparison between different models may be done by the diagnostics tools described in the previous subsection. Additionally, statistical criteria for variable selection may be used, see [43] for a corrected Akaike Information Criterion for GAMLSS. Moreover, boosting is a valuable alternative especially for high-dimensional models [44]. An implementation can be found in the R package `gamboostLSS` (see [45] for a tutorial with examples), yet the set of available distributions is somewhat limited. Here, we retain all variables in the model in order to stay close to the original study.

**4.2.5 Reporting and interpreting the results.** For interpretation purpose it is straightforward to compute marginal effects of the treatment, that is, the change in features of an outcome distribution when the treatment variable changes from 0 to 1 while all other variables are fixed at some specified values. These features may comprise the mean and variance as well as other quantities describing a distribution, such as the Gini coefficient or the vulnerability as expected poverty. The latter we define as the probability of falling below 60% of the median food consumption in our sample (which corresponds to about 95 Pesos). Finally, $t$-tests and confidence intervals can be calculated for testing the presence of marginal effects of the treatment on various measures.

The results in Table 2 show point estimates and 95% bootstrap percentile intervals of marginal effects of the treatment at means for an average household, that is, effects on various distributional measures when the treatment changes from 0 to 1 and the other continuous explanatory variables are fixed at their mean values and categorical variables at their modes. The expected significant positive treatment effect on the mean of the dependent variable is found and can be interpreted as follows: For an average ineligible household, living in a treatment village induces an expected increase in food consumption of about 16.232 pesos per adult equivalent. Although associated with large confidence intervals including zero, the effect on the variances is also positive, indicating a higher variability in the food consumption among the ineligibles in the treatment villages. The Gini coefficient is as well slightly bigger in treatment villages and the confidence intervals do not reject the null hypothesis of equal food consumption inequality (measured by the Gini coefficients) between treatment and control

**Table 2. Treatment effects for ineligibles.**

|  | Estimate | Lower Bound | Upper Bound |
|---|---|---|---|
| ME on mean | 16.232 | 2.833 | 24.685 |
| ME on variance | 8463.007 | -3037.648 | 16159.719 |
| ME on Gini coefficient | 0.014 | -0.009 | 0.036 |
| ME on Atkinson index (e = 1) | 0.012 | -0.008 | 0.033 |
| ME on Atkinson index (e = 2) | 0.018 | -0.010 | 0.051 |
| ME on Theil index | 0.019 | -0.018 | 0.055 |
| ME on vulnerability | -0.015 | -0.048 | 0.009 |

Notes: Shown are point estimates for marginal effects of the treatment at means (ME) and corresponding 95% bootstrap confidence interval bounds based on 499 bootstrap replicates. $n = 4,248$.

villages. We also report effects on other inequality measures, namely the Atkinson index with inequality parameters e = 1, 2 and the Theil index. The results are qualitatively comparable to the effect on the Gini coefficient. To put it differently: There is no evidence that the treatment decreases inequality for an average household among the ineligibles, even though a positive effect on the average food consumption can be found. Furthermore, vulnerability as expected poverty does not change significantly due to the treatment, yet the point estimate indicates a decrease by -0.015, corresponding to an estimated probability of falling below the poverty line of 0.111 for an average household in a control village and the respective probability of 0.096 for an average household in a treatment village. The findings can be illustrated graphically: Fig 3 shows the estimated conditional food consumption distributions for an average ineligible household living in treatment and control villages: It can be seen that the distribution for the treated household is shifted to the right which corresponds to a higher mean and a lower probability of falling below the poverty line. Moreover, the peak of the mode is somewhat smaller

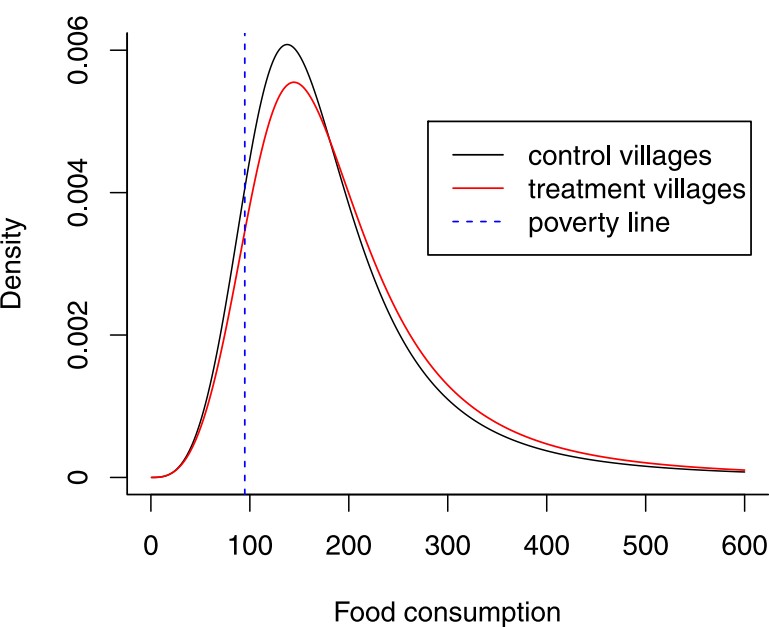

**Fig 3. Estimated conditional distributions for an average ineligible household.**

**Table 3. Treatment effects for eligibles.**

|  | Estimate | Lower Bound | Upper Bound |
|---|---|---|---|
| ME on mean | 28.900 | 16.930 | 35.200 |
| ME on variance | 4550.073 | 837.941 | 7593.226 |
| ME on Gini coefficient | 0.007 | -0.009 | 0.024 |
| ME on Atkinson index (e = 1) | 0.006 | -0.007 | 0.021 |
| ME on Atkinson index (e = 2) | 0.012 | -0.008 | 0.037 |
| ME on Theil index | 0.007 | -0.014 | 0.028 |
| ME on vulnerability | -0.077 | -0.124 | -0.058 |

Notes: Shown are point estimates for marginal effects of the treatment at means (ME) and corresponding 95% bootstrap confidence interval bounds based on 499 bootstrap replicates. $n = 10,492$.

and the right tail in this right-skewed distribution is slightly fatter, resulting in an increased variance and thus higher inequality.

The preceding analyses were conducted for an average ineligible household. Clearly, marginal effects could be obtained for other covariate combinations to investigate how the treatment effect looks like for specific subgroups. Even more heterogeneity can be allowed for by including interactions between the treatment variable and other covariates. In general, we recommend computing marginal effects at interesting and well-understood covariate values rather than averaging marginal effects which mask the heterogeneity of the single marginal effects and could be affected overly strongly by observations that are not of primary interest. However, aggregating marginal effects over all households in the sample is as straightforward as showing the distribution of all these single marginal effects.

Qualitatively the same results emerge for the group of eligibles, as can be seen in Table 3. The treatment effects on the mean are even bigger, still the Gini coefficient and other inequality measures do not decline significantly. In contrast, the point estimates rather indicate a slight increase. A significant decrease is observed for the vulnerability as expected poverty.

Of particular interest are the results on the treatment effects on inequality for all households. In Table 4, we see no significant decline in food consumption inequality for a household with the average characteristics, a quite sobering result for a poverty alleviation program, even though we find evidence for a smaller vulnerability to poverty due to the treatment. The graph of estimated conditional distributions looks similar to Fig 3 but due to including the eligibles, the difference between the distributions is more pronounced (see S1 Fig for the eligibles and

**Table 4. Treatment effects for all people living in treatment villages.**

|  | Estimate | Lower Bound | Upper Bound |
|---|---|---|---|
| ME on mean | 25.900 | 14.730 | 31.130 |
| ME on variance | 4828.316 | 820.841 | 7750.220 |
| ME on Gini coefficient | 0.007 | -0.006 | 0.021 |
| ME on Atkinson index (e = 1) | 0.006 | -0.005 | 0.018 |
| ME on Atkinson index (e = 2) | 0.012 | -0.004 | 0.034 |
| ME on Theil index | 0.007 | -0.012 | 0.026 |
| ME on vulnerability | -0.056 | -0.092 | -0.040 |

Notes: Shown are point estimates for marginal effects of the treatment at means (ME) and corresponding 95% bootstrap confidence interval bounds based on 499 bootstrap replicates. $n = 14,740$.

S2 Fig for all households). However, the reasons for the findings are equivalent: The shift of the distribution to the right due to the treatment lowers the risk of falling below the poverty line. Additionally, while unequal benefits from the treatment increase the variability of the consumption, the right tail of the distribution becomes fatter, preventing an arguably desired decline in inequality.

## 5 Conclusion

This paper introduces GAMLSS as a modeling framework for analyzing treatment effects beyond the mean in various research areas, including economics, which is the focus of this paper, medicine, and epidemiology. Going beyond mean effects is relevant if the evaluator or the researcher is interested in treatment effects on the whole conditional distribution or derived measures that take parameters other than the mean into account. The main advantage of GAMLSS is that they relate each parameter of a distribution and not just the mean to explanatory variables via an additive predictor. Hence, moments such as variance, skewness and kurtosis can be modeled and the treatment effects on them analyzed. GAMLSS provide a broad range of potential distributions which allows researchers to apply more appropriate distributions than the (log-)normal. Furthermore, each distribution parameter's additive predictor can easily incorporate different types of effects such as linear, nonlinear, random, or spatial effects.

To practically demonstrate these advantages, we re-estimated the (mean) regression that Angelucci and De Giorgi [3] applied to evaluate the well-known Progresa program. They found positive treatment effects on poor and non-poor that were larger for the poor (the target group) than for the non-poor. Their findings suggest that the treatment should consequently also decrease inequality within the two groups and within all households. We tested these hypotheses by applying GAMLSS and could not find any evidence for a decline of the conditional Gini coefficient or other inequality measures due to the treatment. An explanation is that the treatment benefited some households distinctly more than others, leading to a higher variance of consumption between households and a higher amount of households having a considerably high consumption. We thus argue that GAMLSS can help to detect interesting treatment effects beyond the mean.

Besides showing the practical relevance of GAMLSS for treatment effect analysis, this paper bridges the methodological gap between GAMLSS in statistics and popular methods used for impact evaluation. While our practical example considers only the case of an RCT, we also develop frameworks for combining GAMLSS with the most popular evaluation approaches including regression discontinuity designs, differences-in-differences, panel data methods, and instrumental variables in the appendix. We show there further how to conduct (cluster robust) inference using the bootstrap. The bootstrap methods proposed in this paper rely on re-estimation of a GAMLSS model for each bootstrap sample. In cases of large datasets and complex models, such approaches are computationally very expensive. The implementation of a computationally more attractive alternative, maybe in the spirit of the score bootstrap method proposed by Kline and Santos [46], is desirable.

## Appendix

### A Combining evaluation methods for non-experimental data and GAMLSS

As demonstrated in Section 4.1, GAMLSS can be used for the analysis of randomized controlled trials, as those are typically handled within the ordinary regression framework. The same applies to difference-in-differences approaches which only include additional regressors,

namely interactions. In the following, we describe how other commonly used evaluation methods and models (see [47] for an overview) can be combined with GAMLSS.

**A.1 GAMLSS and panel data models.** In the evaluation literature, linear panel data models with fixed or random effects seem to be the preferred choice when individuals are observed over time:

$$y_{it} = \beta_0 + \mathbf{x}'_{it}\boldsymbol{\beta}_1 + \alpha_i + \varepsilon_{it}, \quad i = 1, \dots, N, \ t = 1, \dots, T_i. \tag{16}$$

Here, $i$ denotes the individual and $t$ the time period. The vector of explanatory variables $\mathbf{x}_{it}$ may include a treatment effect of interest, time dummies and control variables. In order to capture unobserved time-invariant factors that affect $y_{it}$, individual-specific effects $\alpha_i$ are incorporated in the model. Commonly, these are modeled as fixed effects if the random effects assumption of independence between the time-invariant effects and the explanatory variables is presumed to fail. The Hausman test is an occasionally used tool to underpin the decision for using fixed effects. Another approach which loosens the independence assumptions was proposed by Mundlak [48]. The idea is to extend the random effects model such that for each explanatory variable which is suspected to be correlated with the random effects, a variable including individual-specific means of that variable is added. If this procedure is done for all explanatory variables, we obtain the model

$$y_{it} = \beta_0 + \mathbf{x}'_{it}\boldsymbol{\beta}_1 + \bar{\mathbf{x}}'_i\boldsymbol{\delta}_1 + \alpha_i + \varepsilon_{it}, \quad i = 1, \dots, N, \ t = 1, \dots, T_i, \tag{17}$$

where $\alpha_i, i = 1, \dots, N$, are random effects, $\bar{\mathbf{x}}_i$ is a vector containing the means of the explanatory variables over all $T_i$ time periods for individual $i$, and $\boldsymbol{\delta}_1$ is the vector of associated coefficients. In this specification, the other vector of coefficients $\boldsymbol{\beta}_1$ only includes the effects of the explanatory variables stemming from their variation around the individual-specific means. Hence, $\boldsymbol{\beta}_1$ in (17) is equivalent to $\boldsymbol{\beta}_1$ in a fixed effects model according to (16).

For nonlinear (additive) panel data models, the same question about the validity of the independence assumption between the random effects and the explanatory variables arises. One can allow for dependence via the Mundlak formulation in the same fashion as described above for linear models, that is, avoiding the explicit inclusion of fixed effects while loosening the independence assumption, see chapter 15 in [49] for more details. As random effects are an integrated part of the GAMLSS framework, GAMLSS specifications can be easily used to model panel data. Assume that $y_{it}$ follows a distribution that can be described by a parametric density $p(y_{it}|\vartheta_{it1}, \dots, \vartheta_{itK})$ where $\vartheta_{it1}, \dots, \vartheta_{itK}$, are $K$ different parameters of the distribution. Then, according to model (17), we can specify for each of these parameters an equation of the form

$$g_k(\vartheta_{itk}) = \beta_0^{\vartheta_k} + \mathbf{x}'_{it}\boldsymbol{\beta}_1^{\vartheta_k} + \bar{\mathbf{x}}'_i\boldsymbol{\delta}_1^{\vartheta_k} + \alpha_i^{\vartheta_k}, \quad i = 1, \dots, N, \ t = 1, \dots, T_i, \tag{18}$$

with link function $g_k$, see Sections 2.1 and 2.2 in the main text for details and extensions.

**A.2 Instrumental variables.** Instrumental variable (IV) regression aims at solving the problem of endogeneity bias, for example arising from omitted variables. In this view, an explanatory variable is endogenous, if an unobserved confounder influences the response and is associated with this endogenous variable. That is, we consider the regression specification

$$y = \beta_0 + x_e\beta_e + x_o\beta_o + x_u\beta_u + \varepsilon \quad \text{with} \quad E(\varepsilon|x_e, x_o, x_u) = 0, \tag{19}$$

where $x_o$ is an observed explanatory variable, $x_e$ the endogenous variable, $x_u$ the unobserved confounder, $\varepsilon$ is an error term and $\beta_o, \beta_e$, and $\beta_u$ represent regression coefficients for the observed, endogenous, and unobserved explanatory variable, respectively. However, $x_u$ cannot be observed and thus cannot be included in the model. As $x_u$ is correlated with $x_e$, this violates

the assumption that the error term's expectation given all observed variables is zero. As a consequence, the OLS estimator for $\beta_e$ is inconsistent. In order to demonstrate how a suitable instrument can be used to solve this problem in a nonlinear context, we present the approaches developed for generalized linear models (GLM, [50]), and generalized additive models (GAM, [51] and extend them to the GAMLSS context.

**A.2.1 Instrumental variables in generalized linear models (GLM):** Terza et al. [50] proposed a two-stage residual inclusion procedure (2SRI) that addresses endogeneity in nonlinear models. In fact, the procedure was already suggested by Heckman [52] as a means to test for endogeneity. The reason why ordinary two-stage least squares does not work in the nonlinear context is that the expectation of the response variable is associated via a nonlinear function—the link function in GLMs—with the predictor. Due to this function, the unobserved part is not additively separable from the predictor [51, 53].

In a GLM framework, we consider the model

$$E(\mathbf{y}|\mathbf{X}_e, \mathbf{X}_o, \mathbf{X}_u) = h(\mathbf{X}_e\boldsymbol{\beta}_e + \mathbf{X}_o\boldsymbol{\beta}_o + \mathbf{X}_u\boldsymbol{\beta}_u), \tag{20}$$

where $\mathbf{y}$ is the outcome variable dependent on $\mathbf{X}_o$, a $n \times S_o$ matrix of observed variables, on $\mathbf{X}_e$, a $n \times S_e$ matrix of endogenous variables, and on $\mathbf{X}_u$ which is a $n \times S_u$ vector of unobserved confounders that are correlated with $\mathbf{X}_e$. Consequently, $\boldsymbol{\beta}_o$ is a $S_o \times 1$ vector, $\boldsymbol{\beta}_e$ a $S_e \times 1$ and $\boldsymbol{\beta}_u$ a $S_u \times 1$ vector of regression coefficients. The function $h(\cdot)$ denotes the response function, or the inverse of the link function.

The model in (20) can be written as

$$\mathbf{y} = h(\mathbf{X}_e\boldsymbol{\beta}_e + \mathbf{X}_o\boldsymbol{\beta}_o + \mathbf{X}_u\boldsymbol{\beta}_u) + \boldsymbol{\varepsilon} \tag{21}$$

where the error term $\boldsymbol{\varepsilon}$ is defined as $\boldsymbol{\varepsilon} = \mathbf{y} - h(\mathbf{X}_e\boldsymbol{\beta}_e + \mathbf{X}_o\boldsymbol{\beta}_o + \mathbf{X}_u\boldsymbol{\beta}_u)$ such that

$$E(\boldsymbol{\varepsilon}|\mathbf{X}_e, \mathbf{X}_o, \mathbf{X}_u) = \mathbf{0}. \tag{22}$$

The correlation between $\mathbf{X}_e$ and $\mathbf{X}_u$ is the core of the endogeneity issue at hand. If we were able to observe $\mathbf{X}_u$, consistent estimators for the coefficients in Eq (21) could, for example, be obtained via maximum likelihood estimation (under the usual generalized linear model regularity conditions). Without addressing the endogeneity problem, the $\mathbf{X}_u$ would be captured by the error term leading to a correlation between the explanatory variables and the error.

As in the linear case, to tackle this endogeneity problem, we have to find some observed instrumental variables $\mathbf{W}$ that account for the unobserved confounders $\mathbf{X}_u$. The endogenous variables can be related to these instruments and the observed explanatory variables by a set of auxiliary equations

$$\mathbf{x}_{es} = h_s(\mathbf{X}_o\boldsymbol{\alpha}_{os} + \mathbf{W}_s\boldsymbol{\alpha}_{ws}) + \boldsymbol{\xi}_{us}, \qquad s = 1, \ldots, S_e \tag{23}$$

where $\mathbf{x}_{es}$ is the $s$-th column vector of $\mathbf{X}_e$, $h_s(\cdot)$ is the response function, $\mathbf{W}_s$ is a $n \times S_{IV_s}$ matrix of IVs available for $\mathbf{x}_{es}$ and $\boldsymbol{\alpha}_{os}$ and $\boldsymbol{\alpha}_{ws}$ are $S_o \times 1$ and $S_{IV_s} \times 1$ vectors, respectively, of unknown coefficients. The number of elements in $\mathbf{W}$ must be equal or greater than the numbers of endogenous regressors and there is at least one instrument in $\mathbf{W}$ for each endogenous regressor. The error term $\boldsymbol{\xi}_{us}$ in this model contains information about the unobserved confounders.

The instrumental variables $\mathbf{W}_s$ in Eq (23) have to fulfill the following conditions:

(a). being associated with $\mathbf{x}_{es}$ conditional on $\mathbf{X}_o$

(b). being independent of the response variable $\mathbf{y}$ conditional on the other covariates and the unobserved confounders in the true model, that is, $\mathbf{X}_o, \mathbf{X}_e, \mathbf{X}_u$

(c). being independent of the unobserved confounders $\mathbf{X}_u$.

Terza et al. [50] propose the following procedure to estimate the models in Eqs (21) and (23):

(a). First stage: Get the estimates $\hat{\boldsymbol{\alpha}}_{os}$ and $\hat{\boldsymbol{\alpha}}_{ws}$ for $s = 1, \ldots, S_e$ from the auxiliary Eq (23) via a consistent estimation strategy. One could use maximum likelihood estimation for GLMs here, but nonlinear least squares is also possible. Define

$$\hat{\boldsymbol{\xi}}_{us} = \mathbf{x}_{es} - h(\mathbf{X}_o \hat{\boldsymbol{\alpha}}_{os} + \mathbf{W}_s \hat{\boldsymbol{\alpha}}_{ws}) \qquad \text{for} \quad s = 1, \ldots, S_e. \tag{24}$$

(b). Second stage: Estimate $\hat{\boldsymbol{\beta}}_e, \hat{\boldsymbol{\beta}}_o, \hat{\boldsymbol{\beta}}_{\hat{\Xi}_u}$ via a GLM or a nonlinear least squares method from

$$E(\mathbf{y}|\mathbf{X}_e, \mathbf{X}_o, \hat{\boldsymbol{\Xi}}_u) = h(\mathbf{X}_e \boldsymbol{\beta}_e + \mathbf{X}_o \boldsymbol{\beta}_o + \hat{\boldsymbol{\Xi}}_u \boldsymbol{\beta}_{\hat{\Xi}_u}), \tag{25}$$

where $\hat{\boldsymbol{\Xi}}_u$ is a matrix containing $\hat{\boldsymbol{\xi}}_{us}$ from the first stage as column vectors.

The intuition behind this procedure is that $\hat{\boldsymbol{\Xi}}_u$ contains information on the unobserved confounders if the instruments fulfill the above mentioned requirements. Though $\hat{\boldsymbol{\Xi}}_u$ is not an estimate for the effect of the unobserved confounder on the response variable, its contained information can be used to get corrected estimates for the endogenous variable. Since we are eventually interested in $\boldsymbol{\beta}_e$ and not $\boldsymbol{\beta}_u$, we only need the $\hat{\boldsymbol{\Xi}}_u$ as a quantity containing information about $\mathbf{X}_u$ to account for the presence of these unobserved confounders [51].

**A.2.2 Instrumental variables in generalized additive models (GAM)**: Marra and Radice [51] extend the 2SRI approach to also cover generalized additive models, that allow for nonlinear effects of the explanatory variables on the response variable. A generalized additive model has the following form

$$\mathbf{y} = h(\boldsymbol{\eta}) + \boldsymbol{\varepsilon}, \qquad E(\boldsymbol{\varepsilon}|\mathbf{X}_e, \mathbf{X}_o, \mathbf{X}_u) = 0, \tag{26}$$

where $\mathbf{X}_e = (\mathbf{X}_e^*, \mathbf{X}_e^+)$, $\mathbf{X}_o = (\mathbf{X}_o^*, \mathbf{X}_o^+)$, and $\mathbf{X}_u = (\mathbf{X}_u^*, \mathbf{X}_u^+)$ with matrices containing discrete variables denoted by $*$ and continuous ones by $+$. We summarize the discrete parts of the explanatory variables $\mathbf{X}_e$, $\mathbf{X}_o$, and $\mathbf{X}_u$ into $\mathbf{X}^*$ and the continuous parts into $\mathbf{X}^+$, that is, $\mathbf{X}^* = (\mathbf{X}_e^*, \mathbf{X}_o^*, \mathbf{X}_u^*)$ for discrete variables and $\mathbf{X}^+ = (\mathbf{X}_e^+, \mathbf{X}_o^+, \mathbf{X}_u^+)$ for continuous variables. The linear predictor $\boldsymbol{\eta}$ is represented by

$$\boldsymbol{\eta} = \mathbf{X}^* \boldsymbol{\beta}^* + \sum_{l=1}^{L} f_l(\mathbf{x}_l^+), \tag{27}$$

where $\boldsymbol{\beta}^*$ is a vector of unknown regression coefficients and $f_l$ are unknown smooth functions of $L$ continuous variables $\mathbf{x}_l^+$. These continuous variables can be modeled, for example, by using penalized splines [23]. Since we cannot observe $\mathbf{X}_u^*$ and $\mathbf{X}_u^+$, we get inconsistent estimates for all regression coefficients. Provided that suitable instrumental variables can be identified, we can model the endogenous variables with the following set of auxiliary regressions

$$\mathbf{x}_{es} = h_s(\mathbf{Z}_s^* \boldsymbol{\alpha}_s^* + \sum_{j=1}^{J_s} f_j(\mathbf{z}_{js}^+)) + \boldsymbol{\xi}_{us}, \tag{28}$$

where $\mathbf{Z}_s^* = (\mathbf{X}_o^*, \mathbf{W}_s^*)$ with corresponding coefficients $\boldsymbol{\alpha}_s^*$ and $\mathbf{Z}_s^+ = (\mathbf{X}_o^+, \mathbf{W}_s^+)$, where $\mathbf{Z}_s^+$ is composed of $\mathbf{z}_{js}^+, j = 1, \ldots, J_s$. Instrumental variables meeting the same requirements mentioned above are again denoted by $\mathbf{W}_s$. The smooth functions $f_j$ for the $J_s$ continuous variables

$\mathbf{z}_{js}^+$ include continuous observed variables and continuous instruments. Despite the notation, $f_l$ in (27) and $f_j$ (28) generally are different functions.

Marra and Radice [51] propose the following procedure for the 2SRI estimator within the generalized additive models context:

(a). First stage: Get estimates of $\boldsymbol{\alpha}_s^*$ and $f_j$ for $s = 1, \ldots, S_e$ from the auxiliary Eq (28) using a GAM method. Define

$$\hat{\boldsymbol{\xi}}_{us} = \mathbf{x}_{es} - h_s\left(\mathbf{Z}_s^* \hat{\boldsymbol{\alpha}}_s^* + \sum_{j=1}^{J_s} \hat{f}_j(\mathbf{z}_{js}^+)\right) \qquad \text{for} \quad s = 1, \ldots, S_e. \tag{29}$$

(b). Second stage: Estimate

$$E(\mathbf{y}|\mathbf{X}_e, \mathbf{X}_o, \hat{\boldsymbol{\Xi}}_u) = h_s\left(\mathbf{X}_e^* \boldsymbol{\beta}_e^* + \mathbf{X}_o^* \boldsymbol{\beta}_o^* + \sum_{j=1}^{J} f_j(\mathbf{x}_{jeo}^+) + \sum_{s=1}^{S_e} f_s(\hat{\boldsymbol{\xi}}_{us})\right), \tag{30}$$

where $\mathbf{x}_{jeo}^+, j = 1, \ldots, J$, are column vectors of $\mathbf{X}_{eo}^+ = (\mathbf{X}_e^+, \mathbf{X}_o^+)$.

In this procedure, $f_s(\hat{\boldsymbol{\xi}}_{us})$ accounts for the influence of unmeasured confounders $\mathbf{X}_u$, and we get thus consistent estimates for the observed and the endogenous variables. The set of models in (29) and (30) can be fitted by using one of the GAM packages in R, for example. In simulation studies, Marra and Radice [51] show good performance of the estimates if the instruments are strong.

**A.2.3 Instrumental variables and GAMLSS**: The IV estimation procedure for generalized linear models and generalized additive models can now be transferred to the GAMLSS context. In these models, the response $\mathbf{y}$ follows a parametric distribution with $K$ distributional parameters $\boldsymbol{\vartheta} = (\vartheta_1, \ldots, \vartheta_K)'$ and density

$$p(\mathbf{y}|\mathbf{X}_o, \mathbf{X}_e, \mathbf{X}_u) = p(\mathbf{y}|\vartheta(\mathbf{X}_o, \mathbf{X}_e, \mathbf{X}_u)) \tag{31}$$

For each of the parameters, a regression specification

$$\vartheta_k = h_k(\eta^{\vartheta_k}) \tag{32}$$

is assumed, where $\eta^{\vartheta_k}$ is the regression predictor. For each of the predictors $\boldsymbol{\eta}^{\vartheta_k}$ considered over all $n$ observations, we assume a semiparametric, additive structure

$$\boldsymbol{\eta}^{\vartheta_k}(\mathbf{X}_o, \mathbf{X}_e, \mathbf{X}_u) = \mathbf{X}^* \boldsymbol{\beta}^{*,\vartheta_k} + \sum_{l=1}^{L} f_l^{\vartheta_k}(\mathbf{x}_l^+) \tag{33}$$

Using the same notation as above, the only difference between the Eqs (27) and (33) is that the predictors are now specific for each of the $K$ parameters of the response distribution. Note that the predictors do not have to include the same variables, though the indexes are dropped here for notational simplicity.

If $\mathbf{X}_e$ and $\mathbf{X}_u$ are correlated, then $\mathbf{X}_e$ is endogenous and estimating (33) without considering $\mathbf{X}_u$ leads to inconsistent estimates due to omitted variable bias.

We propose a similar procedure for GAMLSS as the one Marra and Radice [51] developed for GAMs:

(a). First stage: Same as for the GAM procedure.

(b). Second stage: Instead of a GAM, estimate a GAMLSS with density $p(\mathbf{y}|\mathbf{X}_e, \mathbf{X}_o, \hat{\boldsymbol{\Xi}}_u)$ and predictors

$$\boldsymbol{\eta}^{\vartheta_k} = \mathbf{X}_e^* \boldsymbol{\beta}_e^{*,\vartheta_k} + \mathbf{X}_o^* \boldsymbol{\beta}_o^{*,\vartheta_k} + \sum_{j=1}^{J} f_j^{\vartheta_k}(\mathbf{x}_{jeo}^+) + \sum_{s=1}^{S_e} f_s^{\vartheta_k}(\hat{\boldsymbol{\xi}}_{us}). \tag{34}$$

Wooldridge [54] has shown that the 2SRI estimator can be used to model $p(\mathbf{y}|\mathbf{X}_e, \mathbf{X}_o, \hat{\boldsymbol{\Xi}}_u)$ in the second step once models for $E(\mathbf{x}_{es}|\mathbf{X}_o, \mathbf{W}_s)$, $s = 1, \ldots, S_e$, are estimated and the $\hat{\boldsymbol{\xi}}_{us}$ are calculated.

To apply Wooldridge's insights to our case, assume we can derive control functions $C_s(\mathbf{X}_o, \mathbf{x}_{es}, \mathbf{W}_s)$, $s = 1, \ldots, S_e$, such that

$$p(\mathbf{X}_u|\mathbf{X}_o, \mathbf{x}_{es}, \mathbf{W}_s) = p(\mathbf{X}_u|C_s(\mathbf{X}_o, \mathbf{x}_{es}, \mathbf{W}_s)). \tag{35}$$

Here, $C_s(\cdot)$ acts as a sufficient statistic to take account of the endogeneity. For example, if

$$\mathbf{x}_{es}|\mathbf{X}_o, \mathbf{W}_s \sim N(\boldsymbol{\eta}^{\vartheta_k}(\mathbf{X}_o, \mathbf{W}_s), \boldsymbol{\sigma}_{es}^2), \tag{36}$$

then

$$\boldsymbol{\xi}_u = \mathbf{x}_{es} - \mathbf{Z}_s^* \boldsymbol{\alpha}_s^* + \sum_{j=1}^{J_s} f_j(\mathbf{z}_{js}^+) \tag{37}$$

is an appropriate control function in the sense that assumption (35) holds. In this case, including the first-stage residuals $\hat{\boldsymbol{\xi}}_u$ in the second stage, as described in the IV procedures above, is justified. The control function approach is also adopted, for instance, in Blundell and Powell [55] for binary responses and continuous regressors. Instead of using splines in the first stage, they rely on simpler kernel estimators but advocated the use of more sophisticated methods.

Assumption (35) does not hold in general if the model for the endogenous variable is non-linear (first stage). However, as Terza et al. [50] and Marra and Radice [51] have shown, 2SRI still works approximately. Wooldridge [54] recommended including $\hat{\boldsymbol{\xi}}_u$ nonlinearly and/or interactions with $\mathbf{X}_e, \mathbf{X}_o$ in (34) to improve the approximation. Furthermore, a simulation study on different 2SRI settings suggested standardizing the variance of the first stage residuals [56].

The procedure's implementation is similar to the previous one. In the first stage, we estimate a GAM model with one of the available software packages and the second stage is estimated using `gamlss`. That is, while in the first stage the expected mean of the endogenous variables conditional on the other explanatory variables and the instruments are modeled, the distributional part comes only into play in the second stage. The reason is that our interest is on the distribution of the response variable and the first stage serves only as an auxiliary model to account for the endogeneity. In similar contexts, when combining two stage IV estimation and expectile regression, Sobotka et al. [57] show in simulations that it is sufficient to focus on the conditional means in the first stage. They also outline a bootstrap procedure that we modify to our case and is presented in Section B.3.

**A.3 Regression discontinuity design.** In the regression discontinuity design (RDD), see, for example, [58] and [59] for introductions, a forcing variable $X_i$ fully (sharp RDD) or partly (fuzzy RDD) determines treatment assignment. We first consider the sharp RDD case and adopt a common notation for the RDD, as used by Imbens and Lemieux [58], for example. Let the treatment variable be $T_i$ which equals 1 if $X_i$ is bigger than some cutoff value $c$ and 0 if

$X_i < c$. Then, one is typically interested in the average treatment effect on the mean at the cutoff value

$$\tau_{\mathrm{SRD}} = \lim_{x \downarrow c} \mathrm{E}[Y_i | X_i = x] - \lim_{x \uparrow c} \mathrm{E}[Y_i | X_i = x], \tag{38}$$

where $Y_i$ is the dependent variable of interest. The two quantities in (38) may be generally estimated by fitting separate regression models for all or a range of data on both sides of the cutoff value and calculating their predictions at the cutoff value. More precisely, the conditional mean functions $\mathrm{E}[Y_i | X_i, X_i > c]$ and $\mathrm{E}[Y_i | X_i, X_i < c]$ are linked to a linear model via a continuous link function (e.g., identity or logit link). Note that the full range of generalized linear models is included in this formulation, so $Y_i$ may be binary, for instance. Hereby, the crucial assumption is the continuity in the counterfactual conditional mean functions $\mathrm{E}[Y_i(0) | X_i = x]$ and $\mathrm{E}[Y_i(1) | X_i = x]$, where $Y_i = Y_i(0)$ if $T_i = 0$ and $Y_i = Y_i(1)$ if $T_i = 1$. Provided that the assumption holds, the limiting values in (38) can be replaced by the conditional mean functions evaluated at the cutoff and differences in the conditional means can solely be attributed to the treatment. Equally reasonable, one can assume continuity in the density functions $p[Y_i(0) | X_i = x]$ and $p[Y_i(1) | X_i = x]$. In this case, estimators from a wide range of models on many other quantities of the distribution of $Y_i$ (aside from the mean) can be identified in the sharp RDD framework. One example is given in Bor et al. [60] who model the hazard rate in a survival regression. Frandsen et al. [61] derive quantile treatment effects within the RDD. Likewise, the toolbox of GAMLSS can be applied in the sharp RDD. More specifically, assume $Y_i$ follows a distribution that can be described by a parametric density $p(Y_i | \vartheta_{i1}, \ldots, \vartheta_{iK})$ where $\vartheta_{i1}, \ldots, \vartheta_{iK}$ are $K$ different parameters of the distribution. Then, in a simple linear model including only the forcing variable, we can specify for each of these parameters an equation of the form

$$g_k(\vartheta_{ik}) = \beta_0^{\vartheta_k} + X_i \beta_1^{\vartheta_k}, \quad i = 1, \ldots, N, \tag{39}$$

on both sides of the cutoff, where $g_k$ is the link function.

The inclusion of further pre-treatment (baseline) covariates into the regression models of choice on both side of the cutoffs has been deemed uncritical, as they are not supposed to change the identification strategy of the treatment effect of interest, (see, e.g., [58, 59]). Rigorous proofs in Calonico et al. [62] confirm that, under quite weak assumptions, it is indeed justified to adjust for covariates for the frequently used local polynomial estimators in the sharp and fuzzy RDD.

As the interest lies in estimating the treatment effect at the cutoff value, one critical question in the RDD is on which data and in which specification the regressions on both sides of the cutoff should be conducted. Global functions using all data typically need more flexibility and include data far from the cutoff, whereas local estimators rely on a smaller sample size and require the choice of an adequate sample. The apparently most popular approaches in the literature, namely those by Calonico et al. [63] and Imbens and Kalyanamaran [64], use local polynomial regression (including the special case of local linear regression) and thus, a restricted sample. The inherent bandwidth choice is done with respect to a minimized MSE of the estimator for the average treatment effect on the mean. Based on this minimization criterion, a cross-validation approach as originally described in Ludwig and Miller [65] and slightly amended in Imbens and Kalyanamaran [64], is a valuable alternative. In principle, such a cross-validation based bandwidth selection may be transferable to a local polynomial GAMLSS. However, if relying on local estimates, we do not propose using one single bandwidth but rather check the variability of the estimates for different bandwidths, as, for instance, done in Imbens and Kalyanamaran [64]. Additional caution is advised with regard to the diminished sample size resulting from local approaches, as the potentially quite complex

GAMLSS require a moderate sample size. In general, we consider global approaches accounting for possibly nonlinear relationships (e.g., via penalized splines) at least as useful complements to local estimators. In any case, we strongly advocate the visual inspection of a scatterplot displaying the forcing and the dependent variable as well as a careful diagnosis for the estimated models, for example based on quantile residuals in the case of GAMLSS.

The extension to a fuzzy RDD, where the treatment variable $T_i$ is only partially determined by the forcing variable $X_i$, requires some new thinking, namely the idea of compliers. Let us again assume that an individual is supposed to get the treatment if its value of the forcing variable $X_i$ is above a certain cutoff $c$. Then, a complier is an individual that complies with the initial treatment assignment, that is, an individual that would not get the treatment if the cutoff was below $X_i$ but that would get the treatment if the cutoff was higher than $X_i$. Commonly, the interest now lies in the average treatment effect (on the mean) at the cutoff value for compliers

$$\tau_{\text{FRD}} = \frac{\lim_{x \downarrow c} \text{E}[Y|X_i = x] - \lim_{x \uparrow c} \text{E}[Y|X_i = x]}{\lim_{x \downarrow c} \text{Pr}(T_i = 1|X_i = x) - \lim_{x \uparrow c} \text{Pr}(T_i = 1|X_i = x)}, \tag{40}$$

where the denominator now includes the probabilities of treatment at both sides near the cutoff. The treatment effect in (40) is identified under the continuity assumption described above for the sharp RDD and two additional assumptions:

(a). The probability of treatment changes discontinuously at the cutoff value.

(b). Individuals with $X_i$ who would have taken the treatment if $X_i < c$ would also take the treatment if $X_i > c$ and vice versa.

The first assumption ensures that the denominator in (40) does not equal zero (in the sharp RDD, the denominator is by design equal to one). The second assumption, often called the monotonicity assumption, implies that the initial treatment assignment does not have an unintended effect. In other words, individuals do not become ineligible for the treatment or discouraged from taking up the treatment exactly by the initial treatment assignment. We refer to Imbens and Lemieux [58] for a detailed discussion on the average causal effect at the cutoff value for compliers.

As in the sharp RDD, assuming the continuity assumption for the density functions $p[Y_i(0)|X_i = x]$ and $p[Y_i(1)|X_i = x]$ to hold, the numerator in (40) may also contain differences in other quantities aside from the conditional means. The probabilities in the denominator in (40) can be estimated separately, for example via a logistic regression of the treatment variable on the forcing variable, see also chapter 21 in [49]. All remaining considerations from the sharp RDD carry over to the fuzzy case, indicating that GAMLSS can be applied both in the sharp and the fuzzy RDD.

## B Bootstrap inference

In the following, we first describe very generic bootstrapping strategies to obtain inferential statements in the GAMLSS context (Section B.1). Peculiarities of the models discussed in this paper are described in the Sections B.2–B.4. Practical recommendations for diagnosing bootstrap estimates are given in B.5.

**B.1 General strategy.** To fix ideas, assume without loss of generality that the quantity of interest is denoted by $\theta$ and represents the marginal effect of the treatment at the means, namely the treatment effect for an average individual on the Gini coefficient. We consider the parametric bootstrap as the natural choice for a parametric model such as a GAMLSS, although a nonparametric bootstrap is possible as well. The parametric bootstrap works as follows:

(a).   A GAMLSS is fitted to the dataset at hand including $n$ observations. Therefore, $n$ estimated distributions for the dependent variable are obtained.

(b).   A bootstrap sample is generated by drawing randomly one number from each of these estimated distributions.

(c).   The GAMLSS from the first step is re-estimated for the current bootstrap sample. For treated and non-treated individuals, the conditional distributions at mean values for other covariates are predicted. For these distributions, the respective Gini coefficients are computed and their difference is calculated. This difference between the coefficients is the estimated marginal effect of the treatment at means on the Gini coefficient and is denoted by $\hat{\theta}_b$ for the current bootstrap sample.

(d).   The two preceding steps are repeated for many times, say $B$ times.

From the resulting $B$ bootstrap estimates $\hat{\theta}_1, \ldots, \hat{\theta}_B$, bootstrap inference can be conducted in different ways. One option is to perform a $t$-test based on the bootstrap variance

$$\hat{V}_{\text{boot}}[\hat{\theta}] = \frac{1}{B-1}\sum_{b=1}^{B}(\hat{\theta}_b - \bar{\hat{\theta}})^2 \qquad (41)$$

with $\bar{\hat{\theta}} = \frac{1}{B}\sum_{b=1}^{B}\hat{\theta}_b$. To test for significance of the marginal effect of the treatment on the Gini, the $t$-statistic

$$t = \frac{\hat{\theta}}{\sqrt{\hat{V}_{\text{boot}}[\hat{\theta}]}} \qquad (42)$$

can be used, where $\hat{\theta}$ may be the estimate for the marginal effect of the treatment from the original sample or the mean of all bootstrap estimates.

Alternatively, a bootstrap percentile confidence interval can be computed. For instance, the bounds of a possibly asymmetric 95% percentile bootstrap confidence interval are given by the lower 2.5th and the upper 97.5th percentile of the $B$ bootstrap estimates, $\hat{\theta}_1, \ldots, \hat{\theta}_B$. Whereas the idea and implementation of such a confidence interval are straightforward, generally more bootstrap samples and thus, more computational power are required than in the case of using bootstrapped standard errors as outlined above. More elaborate bootstrap confidence intervals exist. Efron [66], for example, proposed a bias-corrected and accelerated method that we do not discuss here. We refer to Efron and Tibshirani [67] and Chernick et al. [68] for more details on parametric and nonparametric bootstrap methods as well as on different techniques to derive bootstrap confidence intervals and $p$-values.

**B.2 Bootstrap inference for grouped and panel data.**   For random effects panel data models where individuals are observed over time and more generally for all random effects models where individuals are grouped into clusters, one has to sample the random effects from their assumed distribution in each bootstrap step first. The distributions for the dependent variable for each individual can then be estimated and the bootstrap dependent variables are drawn from the resulting distributions, corresponding to the first two steps described in Section B.1.

A different approach to account for grouping structures are cluster-robust standard errors. Cameron and Miller [69] give a comprehensive overview on cluster-robust inference, also within the bootstrap machinery. As a method also applicable to nonlinear models, they propose a nonparametric pairs cluster bootstrap to obtain cluster-robust inference. Assume again

that the aim is a significance statement on the marginal effect of the treatment at means on the Gini coefficient and that the sample consists of $G$ clusters or groups. Then, repeat the following procedure $B$ times:

(a).   Resample $G$ clusters $(\mathbf{y}_1, \mathbf{X}_1), \ldots, (\mathbf{y}_G, \mathbf{X}_G)$ with replacement from the $G$ clusters in the original sample, where $(\mathbf{y}_g, \mathbf{X}_g)$, $g = 1, \ldots, G$, denote the vector of the dependent variable and the matrix of the explanatory variables, respectively, for cluster $g$.

(b).   Run the GAMLSS for the bootstrap sample obtained in step (a) and predict the respective conditional distributions at mean values for other covariates for treated and non-treated individuals. For these distributions, the respective Gini coefficients are computed and their difference is calculated. This difference between the coefficients is the estimated marginal effect of the treatment at the means on the Gini coefficient and is denoted by $\hat{\theta}_b$ for the current bootstrap sample.

In complete analogy to our elaborations for non-clustered data, a bootstrap $t$-test can be conducted with the denominator in (42) now based on the cluster-robust variance estimator

$$\hat{V}_{\text{clu;boot}}[\hat{\theta}] = \frac{c}{B-1} \sum_{b=1}^{B} (\hat{\theta}_b - \bar{\hat{\theta}})^2, \tag{43}$$

where $\bar{\hat{\theta}} = \frac{1}{B} \sum_{b=1}^{B} \hat{\theta}_b$ and $c = \frac{G}{G-1} \frac{N-1}{N-K}$ is a finite sample modification with the number of estimated model quantities denoted by $K$.

Alternatively, bootstrap percentile confidence intervals and tests can be constructed from the bootstrap estimates, see the explanations in Section B.1.

**B.3 Bootstrap inference for instrumental variables.**   Due to the stepwise approach in IV methods, the estimation uncertainty arising from the first stage has to be accounted for in the second stage. In order to draw inference for IV models, we propose the following procedure:

(a).   Conduct a parametric bootstrap with $N_b$ replications as described in B.1 for the first stage model in Eq (28).

(b).   With $\hat{\boldsymbol{\alpha}}_s^{[k]}, k = 1, \ldots, N_b$, denoting all of the first stage estimates including the estimates for the smooth functions $f$, calculate

$$\hat{\mathbf{x}}_{es}^{[k]} = h(\mathbf{Z}_s \hat{\boldsymbol{\alpha}}_s^{[k]}) \tag{44}$$

and

$$\hat{\boldsymbol{\xi}}_{us}^{[k]} = \mathbf{x}_{es} - \hat{\mathbf{x}}_{es}^{[k]}. \tag{45}$$

(c).   For the distributional model in the second stage, replace $\hat{\boldsymbol{\xi}}_{us}$ with $\hat{\boldsymbol{\xi}}_{us}^{[k]}$ and proceed as in the general parametric bootstrap procedure described in B.1.

As an alternative to the parametric bootstrap in step 1, a nonparametric bootstrap approach can be applied by drawing bootstrap samples from $\mathbf{x}_{es}$ and $\mathbf{Z}_s$ to get estimates $\hat{\boldsymbol{\alpha}}_s^{[k]}$ of the first stage model.

Let the number of replicates in the second stage be $N_d$, yielding a total of $N_b * N_d$ replicates for the estimates of interest in the second stage. This procedure can be computationally costly if $N_b$ or $N_d$ are chosen to be large. See [51] for a computationally more efficient procedure that assumes approximately normally distributed estimators in the first and second stage, respectively.

**B.4 Bootstrap inference for RDD.** Regressions in the sharp RDD require the estimation of two GAMLSS in each bootstrap sample, namely one on each side of the cutoff value. In the fuzzy RDD, each bootstrap step should also include the re-estimation of the models for the probabilities of the treatment assignment which are chosen to estimate the quantities in the denominator in (40). By doing so, the uncertainty of those estimates is included in the resulting standard errors or confidence intervals for the treatment effect of interest.

**B.5 Recommendations for diagnosing bootstrap estimates.** Irrespective of the impact evaluation and bootstrap method chosen, but especially in the case of the pairs cluster bootstrap, a thorough inspection of the estimated bootstrap statistics is advisable. If the resulting distribution contains large outliers, one should carefully contemplate disusing or at least amending the currently applied bootstrap procedure. Cameron and Miller [69] give a more detailed guideline on diagnosing bootstrap estimates. In our example, the distribution of the bootstrap estimates for the marginal effect of the treatment at the means on the Gini does not reveal large outliers or severe skewness, as can be seen in the boxplot and the histogram in S3 Fig.

The question arises of how many bootstrap samples should be generated. Common choices such as $B = 999$ may be applied. Alternatively, inspecting graphically the convergence of the estimated quantities for a growing number of bootstrap samples indicates whether the chosen amount is sufficient. Exemplarily, S4 Fig shows the percentile interval bounds for the marginal effects of the treatment on the Gini in the sample of ineligibles for increasing bootstrap replicates. The chosen bootstrap sample size of $B = 499$ seems to be appropriate as a higher amount of replicates would probably not change the results substantially.

## Supporting information

**S1 Fig. Estimated conditional distributions for an average eligible household.**
(EPS)

**S2 Fig. Estimated conditional distributions for an average household.**
(EPS)

**S3 Fig. Distribution of bootstrap estimates of ME on Gini.**
(EPS)

**S4 Fig. Percentile interval bounds for ME on Gini for increasing bootstrap replicates.**
(EPS)

## Acknowledgments

We thank Marion Krämer, Jörg Langbein, and David McKenzie for helpful comments on an earlier draft and we are grateful for financial support from the Open Access Publication Funds of the University of Goettingen.

## Author Contributions

**Conceptualization:** Maike Hohberg.

**Data curation:** Peter Pütz.

**Formal analysis:** Maike Hohberg, Peter Pütz.

**Funding acquisition:** Thomas Kneib.

**Investigation:** Maike Hohberg.

**Methodology:** Maike Hohberg, Peter Pütz, Thomas Kneib.

**Software:** Peter Pütz.

**Writing – original draft:** Maike Hohberg, Peter Pütz.

**Writing – review & editing:** Thomas Kneib.

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
