## [Decision Letter · Decision Letter 0]

26 Sep 2019

PONE-D-19-23795

Treatment effects beyond the mean using distributional regression: methods and guidance

PLOS ONE

Dear Ms Hohberg,

Thank you for submitting your manuscript to PLOS ONE. After careful consideration, we feel that it has merit but does not fully meet PLOS ONE’s publication criteria as it currently stands. Therefore, we invite you to submit a revised version of the manuscript that addresses the points raised during the review process.

We would appreciate receiving your revised manuscript by Nov 10 2019 11:59PM. To enhance the reproducibility of your results, we recommend that if applicable you deposit your laboratory protocols in protocols.io, where a protocol can be assigned its own identifier (DOI) such that it can be cited independently in the future. For instructions see: http://journals.plos.org/plosone/s/submission-guidelines#loc-laboratory-protocols

We look forward to receiving your revised manuscript.

Kind regards,

Feng Chen

Academic Editor

PLOS ONE

Journal Requirements:

'NO'

Additional Editor Comments (if provided):

Reviewers' comments:

Reviewer's Responses to Questions

**Comments to the Author**

1. Is the manuscript technically sound, and do the data support the conclusions?

Reviewer #1: Yes

Reviewer #2: Partly

2. Has the statistical analysis been performed appropriately and rigorously? 

Reviewer #1: Yes

Reviewer #2: Yes

3. Have the authors made all data underlying the findings in their manuscript fully available?

Reviewer #1: Yes

Reviewer #2: Yes

4. Is the manuscript presented in an intelligible fashion and written in standard English?

Reviewer #1: Yes

Reviewer #2: Yes

5. Review Comments to the Author

Reviewer #1: This paper proposes the GAMLSS for analyzing treatment effects beyond the mean. The paper is well written and easily accessible. However, it is somewhat lengthy. Moreover, as indicated in the Introduction section, the GAMLSS has already been advocated by Rigby and Stasinopoulos (2005). The authors should further highlight their academic contributions of this research: The first work on using the GAMLSS for analyzing treatment effects beyond the mean?

The authors suggested that the proposed GAMLSS is able to account for spatial effects/heterogeneities in the response variable. Spatial autocorrelation is a typical kind of spatial effects, which has been found in many real-world observation data. Accounting for the spatial autocorrelation in the statistical models would help improve model estimation, reduce model misspecification, and avoid misidentification of significant factors. The following is some examples in the field of traffic safety analysis:

Zeng Q., Gu W., Zhang X., Wen H., Lee J., Hao W. (2019). Analyzing freeway crash severity using a Bayesian spatial generalized ordered logit model with conditional autoregressive priors. Accident Analysis & Prevention, 127, 87-95.

Zeng, Q., Guo, Q., Wong, S. C., Wen, H., Huang, H., Pei, X. (2019). Jointly modeling area-level crash rates by severity: A Bayesian multivariate random-parameters spatio-temporal Tobit regression. Transportmetrica A: Transport Science, 15 (2): 1867-1884.

Zeng, Q., Wen, H., Huang, H., Abdel-Aty, M. (2017). A Bayesian spatial random parameters Tobit model for analyzing crash rates on roadway segments. Accident Analysis & Prevention, 100, 37-43.

In the existence of spatial autocorrelation, the response variables become correlated. Is the GAMLSS able to handle this issue? At least, the general framework illustrated in Section 2.1 did not show this ability.

In line 273, some references should be cited on the tobit model, such as:

Zeng, Q., Wen, H., Huang, H., Abdel-Aty, M. (2017). A Bayesian spatial random parameters Tobit model for analyzing crash rates on roadway segments. Accident Analysis & Prevention, 100, 37-43.

Zeng Q., Wen H., Huang H., Pei X., Wong S.C. (2018). Incorporating temporal correlation into a multivariate random parameters Tobit model for modeling crash rate by injury severity. Transportmetrica A: Transport Science, 14 (3): 177-191.

In line 154, Equation (3) should be Equation (4).

Reviewer #2: The topic of this paper is interesting, and the methods used are sound. The quality of this paper is high. There are several suggestions to improve this paper.

1. This paper is a little bit lengthy, so it is not easy to follow. One suggestion is that the comparison of the GAMLSS and previous models could be put together; therefore the readers can easily know the contribution of this paper.

2. As the authors stated, there are several advantages of GAMLSS; for example, it can consider panel data, random effect, discrete and multivariate distributions, and over-dispersion and zero-inflation. There need be some references for the reasons why this consideration is important. For example, the following ones about panel data [1-2], discrete and multivariate distributions [3-4], over-dispersion and zero-inflation [2, 5]

[1] Analysis of hourly crash likelihood using unbalanced panel data mixed logit model and real-time driving environmental big data. JOURNAL OF SAFETY RESEARCH. 2018, 65: 153-159.

[2] Crash Frequency Modeling Using Real-Time Environmental and Traffic Data and Unbalanced Panel Data Models, International Journal of Environmental Research and Public Health, 2016, 13(6), 609. 7

[3] Injury severities of truck drivers in single- and multi-vehicle accidents on rural highway, Accident Analysis and Prevention, 2011, 43(5), 1677-1688.

[4] Investigation on the Injury Severity of Drivers in Rear-End Collisions Between Cars Using a Random Parameters Bivariate Ordered Probit Model, International Journal of Environmental Research and Public Health, 2019, 16(14) , 2632.

[5] Crash Frequency Analysis Using Hurdle Models with Random Effects Considering Short-Term Panel Data”, International Journal of Environmental Research and Public Health, 2016, 13(11) ,1043.

3. For the model estimation methods, maximum likelihood and Bayesian methods, the possible reference [1-5] maybe also be referred.

4. “Figure 3” could be “Fig. 3”

5. For the marginal effects, the authors need to mention which kind of calculation is used, for example, elasticity and so on, which could be referred to [3].

6. PLOS authors have the option to publish the peer review history of their article (what does this mean?). If published, this will include your full peer review and any attached files.

Reviewer #1: No

Reviewer #2: No

---

## [Author Response · Author response to Decision Letter 0]

11 Nov 2019

Please see the attached PDF file "Response to Reviewers".

---

## [Decision Letter · Decision Letter 1]

2 Dec 2019

Treatment effects beyond the mean using distributional regression: methods and guidance

PONE-D-19-23795R1

Dear Dr. Hohberg,

We are pleased to inform you that your manuscript has been judged scientifically suitable for publication and will be formally accepted for publication once it complies with all outstanding technical requirements.

With kind regards,

Feng Chen

Academic Editor

PLOS ONE

Additional Editor Comments (optional):

Reviewers' comments:

Reviewer's Responses to Questions

**Comments to the Author**

1. If the authors have adequately addressed your comments raised in a previous round of review and you feel that this manuscript is now acceptable for publication, you may indicate that here to bypass the “Comments to the Author” section, enter your conflict of interest statement in the “Confidential to Editor” section, and submit your "Accept" recommendation.

Reviewer #1: All comments have been addressed

Reviewer #2: All comments have been addressed

2. Is the manuscript technically sound, and do the data support the conclusions?

Reviewer #1: (No Response)

Reviewer #2: Yes

3. Has the statistical analysis been performed appropriately and rigorously? 

Reviewer #1: (No Response)

Reviewer #2: Yes

4. Have the authors made all data underlying the findings in their manuscript fully available?

Reviewer #1: (No Response)

Reviewer #2: Yes

5. Is the manuscript presented in an intelligible fashion and written in standard English?

Reviewer #1: (No Response)

Reviewer #2: Yes

6. Review Comments to the Author

Reviewer #1: (No Response)

Reviewer #2: (No Response)

7. PLOS authors have the option to publish the peer review history of their article (what does this mean?). If published, this will include your full peer review and any attached files.

Reviewer #1: No

Reviewer #2: No

---

## [Editor Report · Acceptance letter]

9 Jan 2020

PONE-D-19-23795R1

Treatment effects beyond the mean using distributional regression: methods and guidance

Dear Dr. Hohberg:

I am pleased to inform you that your manuscript has been deemed suitable for publication in PLOS ONE. Congratulations! Your manuscript is now with our production department.

With kind regards,

on behalf of

Dr. Feng Chen

Academic Editor

PLOS ONE